ecology, health and disease and epidemiology

metabolic theory, activation energy, parasite, climate change

**Author for correspondence:**
Jessica Ann Phillips
e-mail: jessy.a.phillips@gmail.com

# The effects of phylogeny, habitat and host characteristics on the thermal sensitivity of helminth development

Jessica Ann Phillips[1,2], Juan S. Vargas Soto[1,3], Samraat Pawar[4], Janet Koprivnikar[5], Daniel P. Benesh[6,7] and Péter K. Molnár[1,3]

[1]Department of Ecology and Evolutionary Biology, University of Toronto, Toronto, Ontario, Canada
[2]Department of Zoology, Oxford University, Oxford, UK
[3]Laboratory of Quantitative Global Change Ecology, Department of Biological Sciences, University of Toronto Scarborough, Toronto, Ontario, Canada
[4]Department of Life Sciences, Imperial College London, Silwood Park, Ascot, UK
[5]Department of Chemistry and Biology, Ryerson University, Toronto, Ontario, Canada
[6]Molecular Parasitology, Humboldt University, Philippstr. 13, Haus 14, 10115 Berlin, Germany
[7]Leibniz-Institute of Freshwater Ecology and Inland Fisheries (IGB), Müggelseedamm 310, 12587 Berlin, Germany

JAP, 0000-0003-1373-3791; JSVS, 0000-0003-3279-3132; SP, 0000-0001-8375-5684;
JK, 0000-0001-8410-1041; DPB, 0000-0002-4572-9546; PKM, 0000-0001-7260-2674

Helminth parasites are part of almost every ecosystem, with more than 300 000 species worldwide. Helminth infection dynamics are expected to be altered by climate change, but predicting future changes is difficult owing to lacking thermal sensitivity data for greater than 99.9% of helminth species. Here, we compiled the largest dataset to date on helminth temperature sensitivities and used the Metabolic Theory of Ecology to estimate activation energies (AEs) for parasite developmental rates. The median AE for 129 thermal performance curves was 0.67, similar to non-parasitic animals. Although exceptions existed, related species tended to have similar thermal sensitivities, suggesting some helminth taxa are inherently more affected by rising temperatures than others. Developmental rates were more temperature-sensitive for species from colder habitats than those from warmer habitats, and more temperature sensitive for species in terrestrial than aquatic habitats. AEs did not depend on whether helminth life stages were free-living or within hosts, whether the species infected plants or animals, or whether the species had an endotherm host in its life cycle. The phylogenetic conservatism of AE may facilitate predicting how temperature change affects the development of helminth species for which empirical data are lacking or difficult to obtain.

## 1. Introduction

Parasitic helminths (i.e. acanthocephalans, cestodes, nematodes and trematodes) are an integral part of almost every ecosystem [1] and can be key drivers of the population dynamics of their hosts and of the food webs in which they are embedded [1–4]. Some helminth species also take a heavy toll on human health, such as schistosomiasis or ascariasis [5,6]. Given that the vast majority of helminths have at least one life stage that is free in the environment and/or rely on an ectotherm intermediate host for progression to the next life stage, climate warming is expected to alter helminth-host interactions around the globe [7,8]. Predicting such impacts is critical for proactive ecosystem management and public health planning [9], but such endeavours are complicated by a lack of data on the thermal sensitivity of greater than 99.9% of all helminth species [10]. Broad generalizations may be possible but are difficult, given that (i) increasing temperatures affect different life-history traits in different ways (e.g. speeding up development but reducing survival); (ii) the thermal sensitivities of life-history traits and host–parasite interactions typically vary with environmental factors, such as the mean and range of temperatures

experienced; (iii) life-history strategies vary substantially among helminth taxa (e.g. directly versus indirectly transmitted nematodes; [11]) and (iv) existing temperature-sensitivity measurements were often reported using idiosyncratic methodologies and metrics [10].

One promising approach for describing, synthesizing, and comparing the thermal sensitivities of helminth species within a common framework is provided by the Metabolic Theory of Ecology (MTE) [9]. The MTE suggests that temperature-sensitivities of life-history traits, such as development rate, are ultimately governed by the temperature-dependence of organismal metabolism and that the thermal sensitivities of more complex ecological interactions can be derived from this basis [10,12]. The exponential increase in the activity levels of metabolic rates within a species' 'Operational Temperature Range' (the range of temperatures typically experienced by individuals of that species [13]) can be captured by the Boltzmann–Arrhenius (BA) model, and the same holds for life-history traits that are directly related to metabolism, such as development rate:

$$y(T) = y_0 \, e^{- E_y/k \, ((1/T) - (1/T_0))}, \tag{1.1}$$

where $y$ is development rate, $T$ is temperature, $y_0$ is the development rate at a reference temperature, $T_0$ and $k$ is Boltzmann's constant. The parameter $E_y$ is the activation energy (AE) of the rate-limiting enzymes (units of electron volts, eV), and describes how steeply metabolic rate, and thus, development rate increases with increasing temperature [12]. At the low and high extremes of an individual's thermal tolerance range, however, enzymes become deactivated, and development first slows and then stops, which, assuming a single rate-limiting enzyme, can be captured by the Sharpe–Schoolfield (SS) equation [14]:

$$y(T) = y_0 e^{-E_y/k(1/T)-(1/T_0))} \cdot (1 + e^{E_y^L/k((1/T_y^L)-(1/T))} + e^{E_y^H/k((1/T_y^H)-(1/T))})^{-1}, \tag{1.2}$$

where parameters are as in equation (1.1), but with the inactivation energies, $E_y^L$ and $E_y^H$, now describing how steeply enzyme activity, and thus, development rate drops to zero around the low and high-temperature thresholds, $T_y^L$ and $T_y^H$.

Understanding how key parameters of equations (1.1) and (1.2) vary among helminth taxa owing to differences in thermal habitat and/or specific characteristics of the host–parasite interactions could substantially facilitate climate change impact predictions, particularly for data-scarce species [10,11]. Powerful commonalities—such as the dependence of the activation energy $E$ on trait function [15], organism type (e.g. lower AEs in plant mortality versus animal mortality [16]), phylogeny [17–19], thermal habitat [15] and proxies thereof (e.g. latitude; [20,21])—have been documented for many non-parasitic species, leading to ecological insights [12] as well as to broad-scale predictions of the impacts of climate warming on ectotherms [22]. Similar commonalities are expected in helminths [23]. However, quantitative reviews of helminth thermal performance have largely focused on the emergence, infectivity, and survival of the free-living, mostly aquatic transmission stages of trematodes (i.e. miracidia and cercaria) [24–27], and their thermal sensitivity need not reflect that of all helminths. Helminths experience diverse thermal habitats, often within a life cycle (e.g. transmission from ectotherm intermediate hosts to endotherm definitive hosts; [11]), and their thermal sensitivities might vary accordingly. Systematic comparisons across helminth taxa within a common framework are lacking.

Here, to our knowledge, we compiled the largest dataset on helminth thermal performance to date, and evaluated whether the thermal sensitivity of development varied with phylogeny, characteristics of the host–parasite system, and/or environmental features.

## 2. Methods

### (a) Literature search and data collection

We searched Web of Science for experimental studies on the temperature-dependence of helminth development, mortality, infection and/or reproduction, published from 1945 to 29 August 2018. We used the following search terms: (helminth or flatworm* or cestode* or tapeworm* or trematode* or fluke* or roundworm* or nematode* or acanthocephalan) and (temperature) and (development or mortality or infection or reproduction or metabolic or survival). We also manually scanned the reference lists of relevant papers, mined a helminth life cycle database [28], and conferred with other researchers to identify additional studies. While we originally aimed to quantify the thermal sensitivity of as many parasite life cycle stages and processes as possible—including survival, infection and reproduction—limited data and complicated host–parasite interaction systems prevented an estimation of the thermal sensitivities of reproduction and host infection in most experiments. Estimates of the thermal sensitivity of mortality were also difficult and associated with more uncertainty than those for development owing to a lower number of experiments, low-temperature resolution in many experiments, and a focus on high temperatures that masks the unimodal nature of survival curves [10]. Therefore, we excluded experiments that focused on host infection or parasite reproduction from further analyses and report estimates for the thermal sensitivity of mortality rates of individual species in the electronic supplementary material, appendix S3 without performing any comparative analyses on data for these traits. We thus focus our analyses on the AEs of developmental processes from here on.

We extracted development rates either directly from a manuscript's text or tables, or if unreported, from its figures using the program GRAPHCLICK [29]. Studies varied in how they measured developmental and other rates, with some, for example, recording the minimum development time from one life stage to the next in an experimental cohort, and others reporting the maximum. Studies also varied in whether they reported rates or times to an event and the chosen units for the reported metrics. As such, we converted each study's data to rates per day and recorded the study's reporting metric as a potential covariate.

For each experiment, we determined the following covariates that may influence temperature sensitivity:

(i) sampling location: if a study reported a specific location from which the experimental parasites were sourced, we recorded the location's latitude, longitude, and altitude using Google EARTH PRO (v. 7.3.1). If no location was reported, we requested the information from the author of the paper where possible. Otherwise, we recorded the location of the lead author's institution as a rough proxy for a possible sourcing location, and evaluated statistically whether this choice influenced our results (cf. below);

(ii) ambient temperature at sampling location: for each location, we obtained the annual mean temperature, the maximum temperature of the warmest month, and the minimum temperature of the coldest month from 1970 to 2000 from WorldClim (http://worldclim.org/version2), which provides temperatures at a spatial resolution of $10 \times 10$ min (roughly 340 km$^2$);

(iii) parasite habitat: we categorized parasites as terrestrial or aquatic. For parasites with complex life cycles and more than one life stage, we considered the habitat of each stage separately;

(iv) geographical distribution: we categorized parasites as 'mostly polar', 'mostly temperate', 'mostly tropical' or 'global', based on distribution data from published papers, helminth parasite guides and online sources such as the Natural History Museum of London host–parasite database [30]. The broad latitudinal distribution was categorized by using a simplified version of updated Köppen–Geiger climate classification maps [31]: species were classified as polar if currently found at latitudes greater than 60° (north or south), temperate if found between latitudes 30°–60° (north or south), tropical if found between 30° north and 30° south, and global if there were documented occurrences in more than one of these sub-categories. We considered the current rather than the ancestral distribution of parasites because this probably reflects their potential thermal tolerance, and determining the endemic range would not be possible for most species;

(v) parasite life cycle: we recorded the life stage at the beginning and end of each development experiment (e.g. first stage larva to third stage larva);

(vi) free-living or in host: we categorized experiments by whether the focal parasite stage was free-living in the environment (e.g. eggs of many species) or in a host (e.g. larvae in intermediate hosts). This determines whether the parasite stage is directly exposed to environmental temperatures or not, and thus, potentially also the thermal sensitivity of biological processes; and

(vii) type of host: we categorized helminth species as plant or animal parasites, and by whether they used an endotherm host or not at some point during their life cycle.

## (b) Estimation of activation energies

We included experiments if they reported non-zero development rates for at least four distinct temperatures (the minimum for fitting the two free parameters of the BA model, equation (1.1)) spanning at least 5°C (thus discarding experiments that tend to be uninformative regarding an organism's temperature sensitivity [13]). For each experiment, we excluded temperature treatments where parasites never completed development because the lack of raw data prevented us from distinguishing between zero development and slow development. We then fitted the BA model (equation (1.1)) to each set of temperature-dependent development rates to estimate the activation energy $E$ using lognormal error distribution, following the procedure outlined in [10]. For experiments that had non-zero development rates from five or more distinct temperatures, we also fitted the SS model, including a high temperature inactivation term (i.e. a unimodal thermal development curve) but ignoring the low-temperature inactivation term in equation (1.2) (owing to the difficulties of detecting and estimating lower temperature thresholds with little data [10,13]).

We used likelihood ratio tests to determine for each experiment whether the data were better fitted by the BA or SS model, but found that the SS fits were often considered better even in the absence of clear thermal optima (e.g. electronic supplementary material, appendix S2: figure S91). The additional parameters of the SS model seemed to facilitate overfitting to slight deviations from the BA curve, so we only used AE estimates from the SS model when there was a clear thermal optimum with more than one measurement beyond this peak. For all other curves, we used the AE estimate from the BA model.

All curve fitting and comparative analyses were conducted in R [32].

## (c) Comparative analysis of temperature dependence

For phylogenetic comparative analyses, we constructed a tree with mitochondrial and ribosomal DNA sequences obtained from Gen-Bank using the most recent classification for each helminth. We aligned sequences using the multiple sequence alignment software MAFFT v. 7 [33], trimmed to eliminate gaps present in more than 15% of the species, and then constructed a maximum-likelihood tree with IQ-TREE [34]. We scaled consensus trees for each helminth group to be ultrametric and then combined them using the divergence times between helminth taxa given in Timetree.org [35]. We added the species in our data without sequences in GenBank to the tree based on taxonomy.

To explore which factors impact the AE of helminth development, we began by fitting phylogenetic mixed models [36,37] with the R package MCMCglmm [38]. To do this, we first weighted AE estimates by their quality, using the squared standard error for curve fitting, thus upweighting estimates from curves with more temperatures and better fits. As some parasite species had multiple AE estimates (e.g. from different life stages or multiple studies), we further included a random effect for parasite species. We also included the phylogenetic covariance matrix as a random effect to account for similarities between related species. A few experiments ($n = 8$) reported multiple summary metrics for the development times observed in a cohort, such as the minimum, mean and/or maximum times spent between one life stage and the next. Given that these experiments revealed systematic differences in AE depending on the metric that was used (electronic supplementary material, appendix S1 and figure S1), we only used the AE associated with the median developmental time in cases where multiple metrics were available, and added measurement metric (min, mean/median, max) to the model as a fixed predictor when analysing the complete dataset.

Next, we sequentially added environmental variables, starting with the sampling location's mean temperature, followed by the range in monthly mean temperatures. We then added latitude and tested if its effect depended on species being categorized as globally distributed (i.e. occurring in multiple climate zones). We checked whether patterns differed when collection locations were known versus inferred from the author's institution for both temperature and latitude. We added habitat as a predictor (aquatic versus terrestrial), and also assessed the interaction between habitat and latitude as aquatic habitats could buffer seasonal temperature fluctuations and, thus, modify the effect of latitude. Finally, we examined the influence of host–parasite characteristics, specifically whether AEs differed between parasite stages free in the environment versus within hosts, between parasites using plant versus animal hosts, and between parasites with or without an endotherm (i.e. bird or mammal) host in their life cycle.

As we added terms, we assessed model improvement by examining the deviance information criterion (DIC), the significance of new parameters, and the overall variance explained ($R^2$, calculated according to [39]). Our dataset included six variables of primary interest: temperature, latitude, habitat (terrestrial versus aquatic), parasite life stage within versus outside of a host, plant versus animal parasite, and whether the parasite has an endotherm in its life cycle. We added these variables to the model regardless of whether they improved model fit. We also assessed plausible interactions between these variables. However, to limit overfitting, we only retained interactions if the parameters were significant ($p < 0.05$) or the model fit increased ($\Delta DIC > 3$). The perceived importance of model terms can depend on the sequence in which they are added [40], particularly when terms are correlated (like temperature and latitude), so we assessed each term when added in series, when included in the full model, and when added alone to the base model accounting for phylogeny.

## 3. Results

## (a) Activation energy estimates within and across species

We analysed 129 thermal performance curves for helminth development from 87 species (three acanthocephalans, 62

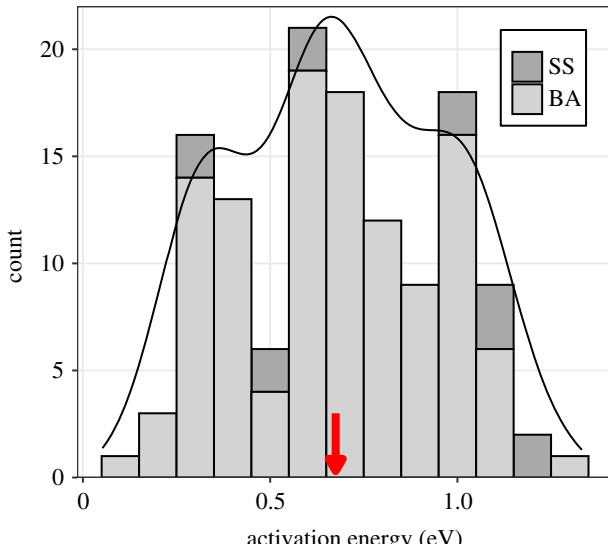

**Figure 1.** Distribution of activation energy (AE) estimates ($n = 129$) for the development rate of 87 helminth species. AE was estimated using the Sharpe-Schoolfield model (SS) when thermal performance curves had an unambiguous peak and using the Boltzmann-Arrhenius model (BA) otherwise. The arrow indicates the median. The solid line is an overall density line, calculated for all AE estimates, regardless of whether they were estimated with the BA or SS model. (Online version in colour.)

nematodes and 22 platyhelminths) based on 4 to 16 temperatures per experiment (median = 6). Only 13 (10%) of the curves had an unambiguous thermal peak such that the SS model was more appropriate than the BA model (see the electronic supplementary material, appendix S2). The median AE was 0.67 electron volts (eV) (interquartile range (IQR): 0.43–0.87 eV) when only estimates from the BA model were considered, 0.99 eV (IQR: 0.51–1.08 eV) when only the 13 estimates from the SS model were considered, and 0.67 eV if BA and SS estimates were considered together (IQR: 0.44 to 0.90 eV; figure 1). These medians were similar to means (0.67, 0.81 and 0.68 eV), indicating limited skew in the distribution of AEs (figure 1). The comparative and statistical results outlined below were qualitatively unaffected by whether SS estimates of AEs were included or excluded from the analyses.

For some helminth species, there were AE estimates from multiple experiments, and these tended to be consistent (within-species random effect, table 1). Adding phylogeny further improved the model (table 1), indicating related species had similar AEs (figure 2). The phylogenetic effect was consistent; the variance explained by random effects (conditional minus marginal $R^2$) did not decrease as fixed predictors were added to the model (table 1). Groups in which development was strongly temperature-dependent included anisakid nematodes (specifically their propagule stages in the external environment) and protostrongylid nematodes (in their snail intermediate hosts), whereas groups with less temperature-sensitive development included hymenolepid tapeworms (in their invertebrate intermediate hosts), and a clade of root-cyst plant parasites (*Globodera*, *Heterodera* and *Rotylenchulus*). AEs differed most among helminth families and orders (electronic supplementary material, appendix S1 and figure S2), and family and order means predicted without other covariates are presented in the electronic supplementary material, appendix S1 and figure S3). Whether developmental time was reported

as a minimum, mean/median, or maximum did not systematically affect AE estimates (table 1).

## (b) Correlates of activation energy

AEs decreased by approximately 0.015 eV (95% credible interval (CI) = −0.028 to −0.002) for each 1°C increase in mean annual temperature (figure 3a) when controlling for other covariates (p = 0.02 in final model; table 1). AE did not clearly vary with temperature range (figure 3b; slope = −0.001 (CI = −0.008 to 0.005), p = 0.70) or latitude (figure 3c; slope = −0.005 (CI = −0.011 to 0.001), p = 0.10). Latitude and temperature effects were not dependent on collection locality being exactly known or on whether species had global distributions (table 1). At an average temperature and latitude, AEs were 0.23 eV (95% CI: 0.008–0.46 eV) (p = 0.04) higher in terrestrial than in aquatic parasites (figure 3d). Together, environmental variables explained approximately 7% of the variation in AE (table 1). AE did not depend on whether the parasite stage in question was inside or outside of a host, whether the parasite infected plants or animals, or whether it had an endotherm in its life cycle (figure 4 and table 1).

## 4. Discussion

We found that the distribution of AEs in helminth parasites is similar to that previously reported for free-living eukaryotic species [15,21], but lower than for prokaryotes [41], with a median AE of 0.67 eV, falling into the 0.6–0.7 eV range. This is important because meaningful thermal performance data exist for less than 0.1% of the estimated hundreds of thousands of extant helminth species [3,10]. Climate change impacts have been projected only for well-studied species, such as *Schistosoma mansoni* (e.g. [42]), but little progress has been made regarding the other 99.9% to date. Given that logistical constraints mean we will never be able to study the thermal sensitivities of every helminth species, and that climate change is outpacing our ability to collect sufficient data for building species-specific projection models for most helminths before impacts occur, there is an urgent need for alternative approaches. Broad, comparative analyses, such as the one we report here, are critical for highlighting commonalities and differences among parasite groups that can inform climate change impact predictions. Moreover, they may also enable predictions for data-scarce species by highlighting similarities with insights from free-living species, systematic relationships based on phylogeny, characteristics of the host-parasite system and/or environmental features, as well as by indicating critical data gaps that need to be filled.

Right-skewed AE distributions have been extensively documented for free-living species [15,21]. Multiple hypotheses have been proposed to explain this skewness, including predator-prey interactions (the 'thermal life-dinner principle'; [15]), unequal sampling across latitudes [21], and effects of fluctuating selection [17]. In helminth parasites, we found limited skewness in the AE distribution (figure 1), and that thermal sensitivities were influenced by certain predictors, but not others. Notably, phylogenetic structure was key for explaining the variation in thermal sensitivity in our dataset. AEs tended to be similar among related species, as well as when measured multiple times from the same species. Such phylogenetic structure is not surprising because similar trends have been noted in free-living

**Table 1.** Mixed models examining activation energies (AEs) for helminth development ($n = 129$; 87 species). (Models were fitted with the Bayesian $R$ package MCMCglmm and compared using the deviance information criterion (DIC). Data points were weighted by the standard error of the AE estimate. Marginal $R^2$ ($R^2_m$) represents the proportion of variation explained by fixed effects, while conditional $R^2$ ($R^2_c$) represents that explained by random and fixed effects combined. For $R^2$ estimates, the 95% posterior credible interval is given.)

| term | d.f. | DIC | $R^2_m$ | $R^2_c$ |
|---|---|---|---|---|
| intercept only | — | 38.0 | 0 | 0 |
| base model | | | | |
| + within-species random effect[a,b] | 1 | −25.0 | 0 | 0.60 [0.37–0.74] |
| + phylogenetic random effect[a,b,c] | 1 | −32.9 | 0 | 0.70 [0.48–0.86] |
| + metric (min, mean, max) | 2 | −28.3 | 0 [0–0.05] | 0.72 [0.48–0.86] |
| environmental covariates | | | | |
| + mean annual temperature[a,c] | 1 | −32.3 | 0 [0–0.07] | 0.68 [0.47–0.85] |
| + mean temp × exact location | 2 | −31.4 | 0.02 [0–0.08] | 0.74 [0.48–0.87] |
| + temperature range | 1 | −34.1 | 0 [0–0.08] | 0.73 [0.49–0.87] |
| + latitude[a] | 1 | −30.1 | 0.02 [0–0.10] | 0.65 [0.48–0.85] |
| + latitude × global distribution | 2 | −33.0 | 0.04 [0.01–0.15] | 0.71 [0.47–0.84] |
| + habitat (terrestrial versus aquatic) [a,b,c] | 1 | −38.3 | 0.07 [0.01–0.22] | 0.82 [0.57–0.92] |
| + habitat × latitude | 1 | −38.3 | 0.08 [0.01–0.23] | 0.83 [0.58–0.92] |
| host–parasite characteristics | | | | |
| + stage inside/outside of host[a] | 1 | −37.2 | 0.09 [0.01–0.23] | 0.81 [0.57–0.92] |
| + plant versus animal host[a] | 1 | −37.4 | 0.09 [0.02–0.28] | 0.83 [0.61–0.93] |
| + endotherm host in cycle versus not[a] | 1 | −40.8 | 0.13 [0.04–0.30] | 0.83 [0.65–0.94] |

[a]Term of biological interest, retained regardless of significance.

[b]Parameter estimated as significant ($p < 0.05$) when added in sequence.

[c]Parameter estimated as significant in the final model.

Note: no parameter was estimated as significant in isolation, i.e. when added to the base model.

taxa [17–19], and because helminth traits as diverse as morphology (e.g. [43]), host specificity (e.g. [44,45]) and population parameters (e.g. [46]) are shaped by phylogeny.

While helminth AEs were correlated with some variables that we used as proxies of the thermal niche, accounting for these correlations did not decrease phylogenetic effects. This suggests that, even with similar thermal niches, helminth clades differed in their temperature sensitivity. Phylogenetic patterns in AE were also not explained by three characteristics of the studied host–parasite systems: infecting plants or animals, being inside or outside of a host, or having an endotherm in the life cycle versus not. Plant and animal helminths did not differ in their AE on average, even though free-living animals tend to have higher AEs than plants [16]. However, this should be interpreted cautiously given the small number of plant parasites in our dataset.

We found clades of parasites that had either high or low AEs regardless of whether the life-history stage of interest was free-living or within a host, or if these were animal parasites with an endotherm in their life cycle. Several nematode clades, often at the family level, had relatively high AEs. One was the family Protostrongylidae (order Rhabditida), with the life-history stage found within terrestrial gastropods. The families Anisakidae and Ascaridiidae within the order Ascaridida represented another clade with high AEs, even though this contained both marine and terrestrial species. Despite this, nematodes did not always have relatively high AEs, with low values seen in the clade represented by plant-infecting species in the orders Tylenchida and

Mermithida regardless of whether they had a tropical or temperate distribution. Similar patterns are known, for example, from studies on host specificity, where some helminth clades also contained species with trait values that varied from most of their relatives [44]. In other words, a clade characterized by high AE values sometimes contained species with low AEs and vice versa, with no obvious explanations for such outliers in either study. The fact that phylogeny was the strongest predictor may thus both help and hinder modelling efforts for understanding parasite responses to altered temperatures. Specifically, species for which thermal sensitivity data are lacking may generally respond in a similar fashion as their relatives, but there can be exceptions within clades.

Phylogenetic structure can arise through non-adaptive processes, like ontogenetic constraints, or through adaptation, e.g. via niche tracking [47]. If temperature sensitivity evolves in response to changes in the thermal niche, one would expect species experiencing different thermal regimes to evolve different AEs. For example, one hypothesis is that environments with high temperature variability favour thermal generalism (low AE), which is, for example, supported by latitudinal gradients in temperature-dependent phytoplankton growth rates [18]. These patterns, though, are not universal [48], with conflicting trends reported both in free-living species (e.g. insects; [20,21]) and helminths (e.g. trematode cercarial emergence; [24,49]). Here, we found that the temperature sensitivity of helminth development did not decrease with two proxies of temperature variability: latitude or the temperature range of the sampling location.

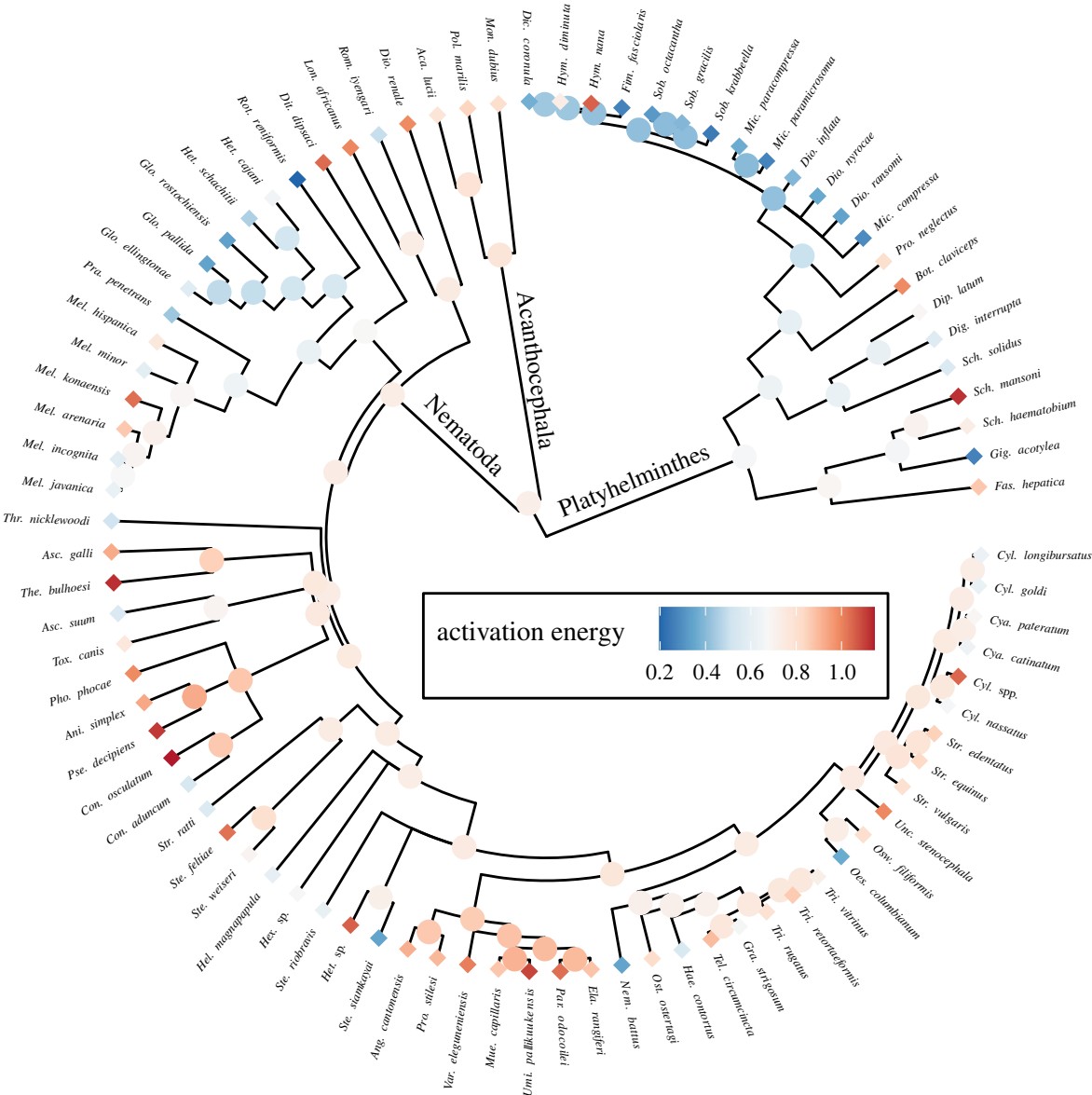

**Figure 2.** Phylogenetic structure in activation energy (AE) estimates for 87 helminth species. AE is colour coded, with tips showing median species values and nodes depicting predictions from a phylogenetic mixed model. (Online version in colour.)

Furthermore, AE was higher in terrestrial than in aquatic helminths, despite presumably greater temperature fluctuations in terrestrial environments (though some aquatic habitats, such as tide pools, may also exhibit high temperature variability). Overall, temperature variability was not associated with lower AE in helminths, but this may also be a consequence of only considering one time scale of fluctuation (during development), while selection on plasticity/generality depends on the time scale of fluctuations relative to the generation time of the organism [17]. Another hypothesis to explain variation in helminth AEs is cold adaptation; species from cold environments should exhibit thermal generalism to maintain trait performance at low temperatures [50]. However, helminths exhibited the opposite trend, with species from colder climates having stronger temperature-dependent development than those from warmer climates. Perhaps it is more beneficial to be able to react quickly to the onset of the shorter growing seasons found in colder regions (high AEs), than to be able to buffer large temperature fluctuations during the growing season with low AEs. Whether helminths from cold, terrestrial habitats somehow benefit from stronger temperature sensitivity requires further investigation.

AEs also did not differ among helminth stages that were inside or outside the host, suggesting that helminth development inside ectotherm hosts does not systematically depend on the thermal sensitivity of host metabolism or immune responses. Helminths with one life stage (generally the adult) living at a stable, high temperature in an endotherm did not exhibit stronger (or weaker) temperature-sensitive development in other life stages, like the eggs in the external environment or larvae in invertebrate hosts. Similarly, different free-living transmission stages of the same helminth, such as trematode miracidia and cercaria, can exhibit varied thermal responses [25]. Other helminth traits, like host specificity, likewise evolve independently in response to different selection pressures on each life stage [44]. Some stages may thus be under stronger selection for thermal generalism than others. Here, hymenolepid tapeworms in arthropod intermediate hosts and nematode lungworms (Protostrongylidae) in snail intermediate hosts exhibited low and high AE, respectively. They also differ in larval developmental times, with the tapeworms developing much faster than the lungworms [51]. Perhaps the conditions favouring rapid helminth development, such as higher

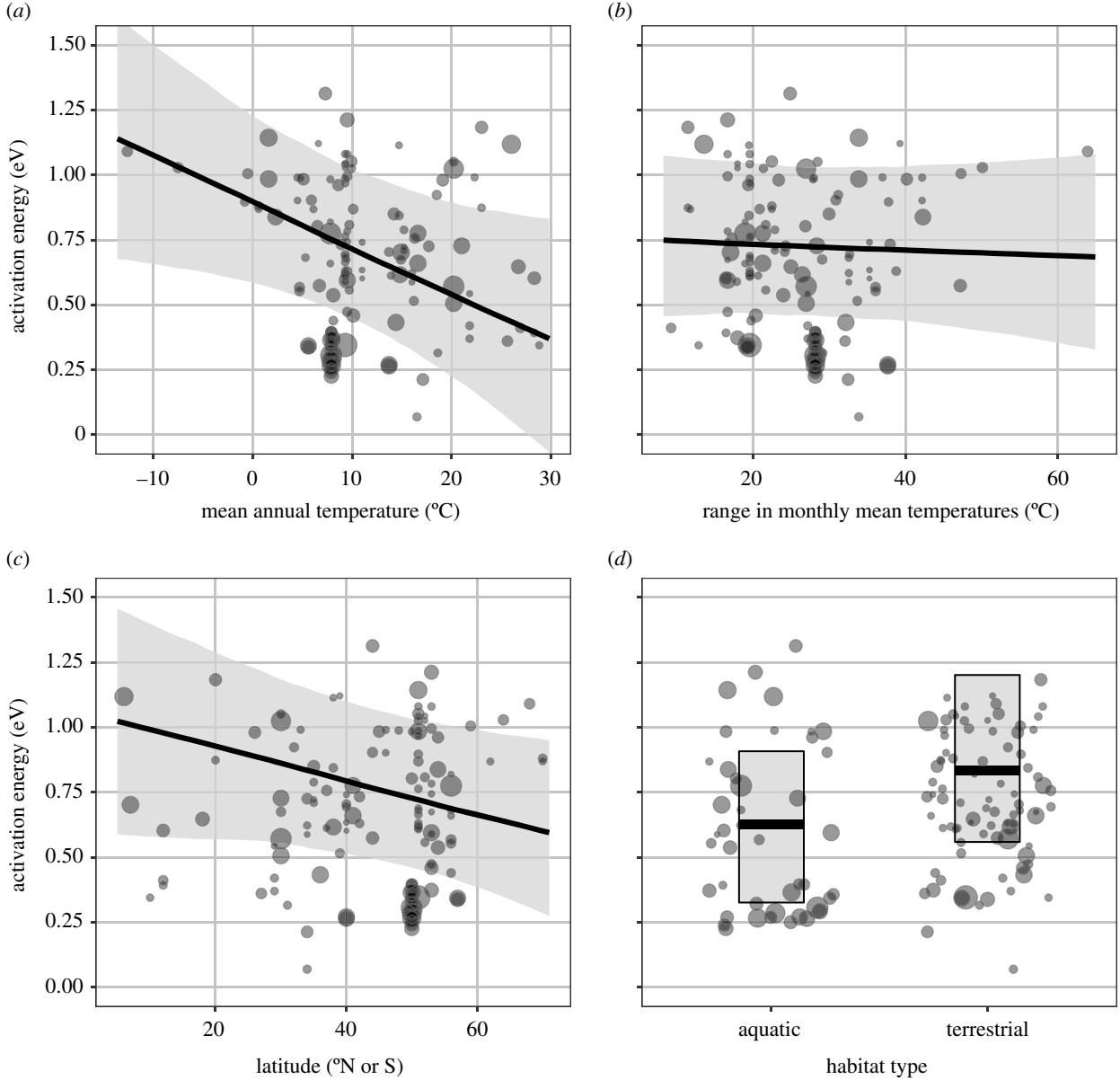

**Figure 3.** Activation energy (AE) of helminth development as a function of four environmental variables: (*a*) mean annual temperature of the parasite's sampling locale, (*b*) range in monthly mean temperatures, (*c*) latitude, and (*d*) habitat. Point diameter is proportional to the inverse standard error of the AE estimate and thus reflects its weight in the analysis. Lines and shaded areas represent best fits and credible intervals from mixed models, including parasite phylogeny and the other environmental variables.

mortality [52], also favour lower thermal sensitivity, analogous to how stronger selection may drive lower thermal sensitivity in prey versus predator performance traits [15].

Our study is limited by the species that have been studied, by what data were and were not reported, and by the methodologies that were used. Many studies, for example, had too small temperature ranges and/or too few temperature treatments to allow full characterization of thermal sensitivities (e.g. using the SS model). While using the BA model for cases where only the intermediate range of a species' thermal niche was captured did not impact our comparative analyses of AEs, it does limit our ability to predict the impacts of warming in cases that involve/are near thermal thresholds (e.g. the onset and end of a transmission season; range expansions and contractions near range edges). There were also idiosyncrasies in which metric studies chose to report (e.g. minimum versus maximum development rates as opposed to the mean/median) and whether development rates may have been influenced by composite processes (e.g. impacts of host immunity on parasite development). However, the biggest

limitation of our study remains that despite compiling over a hundred temperature response curves from studies spanning more than six decades, it only covers a fraction of helminth diversity. Nematodes, for example, are over-represented in our datasets, despite trematodes estimated to represent the largest part of helminth diversity [53]. Broad, systematic and representative samples of parasite diversity and life stages are urgently needed to untangle taxon- and stage-specific differences in parasite temperature sensitivities [53], and reference [10] provides guidance on how to design experiments that capture the full temperature range, avoid pseudoreplication, estimate parameters and separate composite rates. We did not include host phylogeny or body size in this analysis because many helminths have complex life cycles within multi-host systems, and most of our data are restricted to the free-living life stages of parasites. This reflects the difficulty of experimentally measuring traits of helminth parasites across multiple life stages, and especially when they are within hosts. Additional covariates of thermal sensitivity that could not be considered here owing to data limitations, such

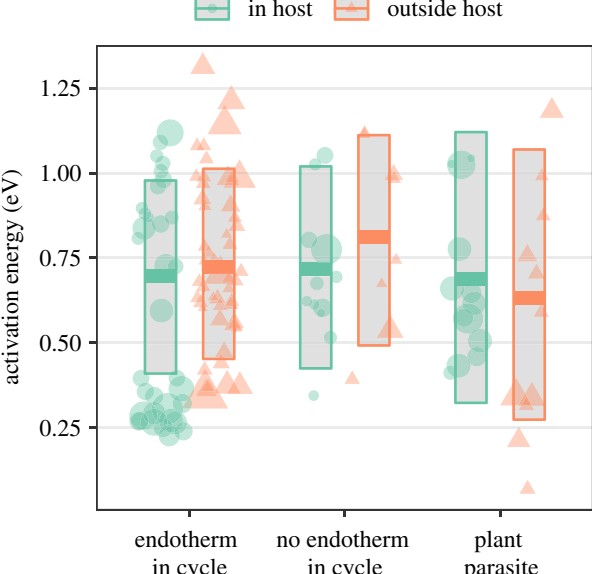

**Figure 4.** Activation energy (AE) of helminth development separated by host types (animal parasites with and without an endotherm in their life cycle and plant parasites) and whether the studied parasite stage was inside a host or outside in the external environment. Point diameter reflects its weight in the analysis, i.e. the inverse standard error of the AE estimate. Lines and shaded areas represent estimated means and credible intervals from mixed models accounting for parasite phylogeny. (Online version in colour.)

as the body size of the parasite stage, the mode of transmission (e.g. active versus passive transmission), and host specificity, should also be considered in such endeavours, for example, by collecting and analysing representative samples of helminth diversity that systematically vary in such factors.

Predicting helminth parasite temperature responses is important for understanding their distribution and spread under projected temperature changes, but remains challenging. This being said, systematic temperature relationships exist and should be exploited in creating general models. Our study highlights such relationships, allowing first guesses at these temperature responses, but also underscoring the many data gaps that exist. Helminth phylogeny was the best predictor of the AE for development, although there was variation within clades, and thermal sensitivity can also vary among life stages of the same species. Nonetheless, phylogeny may be a useful starting point for forecasting how higher temperatures will

affect helminth dynamics in species for which empirical data are lacking or difficult to obtain. Our results suggest that certain helminth taxa will be inherently more affected by warming through effects on their development. Some helminths may benefit from faster development and, thus, easier transmission during constrained seasonal windows, while others will be relatively constrained based on their thermal sensitivity. Moving forward, filling empirical data gaps through systematic sampling, exploring the roles of other drivers, and testing and iteratively improving models based on thermal relationships can lead to a mechanistic predictive framework for climate change impacts on host–parasite dynamics.

Data accessibility. Data and code for calculating activation energies are available from the Dryad Digital Repository: https://doi.org/10.5061/dryad.qbzkh18hm [54]. Code for the phylogenetic comparative analyses are available on GitHub: https://github.com/dbenesh82/helminth_AE_analysis.

Authors' contributions. J.A.P.: data curation, formal analysis, funding acquisition, investigation, writing—original draft, writing—review and editing; J.S.V.S.: formal analysis, investigation, software, visualization, writing—review and editing; S.P.: funding acquisition, investigation, methodology, writing—review and editing; J.K.: data curation, funding acquisition, investigation, writing—original draft, writing—review and editing; D.P.B.: data curation, formal analysis, funding acquisition, investigation, methodology, software, visualization, writing—original draft, writing—review and editing; P.K.M.: conceptualization, formal analysis, funding acquisition, investigation, methodology, project administration, software, supervision, writing—original draft, writing—review and editing.

All authors gave final approval for publication and agreed to be held accountable for the work performed therein.

Competing interests. We declare we have no competing interests.

Funding. J.A.P. was supported by a Rhodes scholarship and a Natural Sciences and Engineering Research Council of Canada (NSERC) Postgraduate Scholarship-Doctoral (grant no. PGSD2-532632-2019). S.P. was supported by Biotechnology and Biological Sciences Research Council (BBSRC) grant no. BB/N013573/1 and National Institutes of Health (NIH) grant no. 1R01AI122284-01 as part of the joint (NIH-NSF-USDA-BBSRC) Ecology and Evolution of Infectious Diseases program. J.K. was supported by a NSERC Discovery grant no. (RGPIN-2020-04622). D.P.B. was supported by the Deutsche Forschungsgemeinschaft (DFG, German Research Foundation)–project number BE 5336/3-1. P.K.M. was supported by a NSERC Discovery grant no. (RGPIN-2016-06301), the Canada Foundation for Innovation (CFI) John R. Evans Leaders Fund (grant no. 35341) and the Ministry of Research, Innovation and Sciences (MRIS) Ontario Research Fund.

Acknowledgements. We thank Hannah O'Sullivan for her assistance with fitting temperature response curves.

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
