## [Peer Review File · Proceedings of the Royal Society B: Biological Sciences]

Review History

RSPB-2021-1878.R0 (Original submission)

Review form: Reviewer 1

Recommendation

Accept with minor revision (please list in comments)

Scientific importance: Is the manuscript an original and important contribution to its field?

Excellent

General interest: Is the paper of sufficient general interest?

Good

Quality of the paper: Is the overall quality of the paper suitable?

Excellent

Is the length of the paper justified?

Yes

Should the paper be seen by a specialist statistical reviewer?

No

Do you have any concerns about statistical analyses in this paper? If so, please specify them explicitly in your report.

No

It is a condition of publication that authors make their supporting data, code and materials available - either as supplementary material or hosted in an external repository. Please rate, if applicable, the supporting data on the following criteria.

Is it accessible?

Yes

Is it clear?

Yes

Is it adequate?

Yes

Do you have any ethical concerns with this paper?

No

Comments to the Author

The effects of phylogeny, habitat, and host characteristics on the thermal sensitivity of helminth development

This topic is both very interesting and relevant, and I am happy to see these authors have compiled such a vast and important dataset for helminths. I think the stronger phylogenetic conservatism supported by their dataset is important (and surprising), as I initially thought parasite traits/ecology would be more important. Overall, the manuscript is well written, and the take home messages are clear.

I only have a few minor suggestions for the authors to consider:

1. I wonder if conducting a comparative analysis of temperature dependences with the addition of host phylogeny is worth considering. It seems like both parasite and host phylogeny would be important and can drive some macroecological patterns (e.g., Harnos et al., 2016 'Size matters for lice on birds: Coevolutionary allometry of host and parasite body size' *Evolution*). Thinking about this further, it could be problematic for multi host lifecycles, so I am unsure what host in the lifecycle is the most appropriate to model.
2. Within host groups (e.g., ectotherms), I suspect host body size could be an important covariate in the models presented here. The idea being bigger hosts can provide some sort of thermal inertia for parasites (similar to the ideas in Bergmann's Rule).

Line 292: I think this sentence should start 'The fact that phylogeny...'

Lines 313-317: This part of the discussion makes me wonder if temperature seasonality (coefficient of variation of temp) would be a useful covariate in the model.

Table 1. Is there a graphical way to represent this information? Maybe consider displaying the estimates of each term in the model. Further, showing the direction of the effects of the covariates would be useful.

Review form: Reviewer 2

Recommendation

Accept with minor revision (please list in comments)

Scientific importance: Is the manuscript an original and important contribution to its field?

Excellent

General interest: Is the paper of sufficient general interest?

Excellent

Quality of the paper: Is the overall quality of the paper suitable?

Good

Is the length of the paper justified?

Yes

Should the paper be seen by a specialist statistical reviewer?

No

Do you have any concerns about statistical analyses in this paper? If so, please specify them explicitly in your report.

No

It is a condition of publication that authors make their supporting data, code and materials available - either as supplementary material or hosted in an external repository. Please rate, if applicable, the supporting data on the following criteria.

Is it accessible?

Yes

Is it clear?

Yes

Is it adequate?

Yes

Do you have any ethical concerns with this paper?

No

Comments to the Author

The authors present a study on the thermal sensitivity of helminth parasites by comparing the activation energies (AEs) for developmental rates based on available data from experimental studies. To my knowledge, this study presents the first approach that analyses the thermal sensitivity of multiple parasite helminth taxa from different phyla in such a unifying framework. Ultimately, such large-scale assessments will be useful to better estimate how different parasite taxa will respond to climate change and global warming pressures. However, as the authors point out, empirical data gaps still hinder current assessments and predictions and will need to be addressed to provide better predictive frameworks for the impacts of climate change on parasite dynamics. Overall, this is a relevant and well-written manuscript that should be of interest to a wide readership.

My main criticism is that the introduction and discussion could be 'synchronised' a bit better. In particular, in the introduction the activation energy AE is only mentioned briefly in the context of the BA model (L58, and as an example in L69), yet throughout the rest of the text AE is used and

discussed as the main indicator of thermal sensitivity in helminth development. In my opinion, AE could therefore be explained a bit more in detail in the introduction for a general readership.

Secondly, the authors conclude that “phylogenetic structure was key for explaining the variation in thermal sensitivity in our dataset” (L267). Was there any overall difference between nematodes, trematodes, cestodes or acanthocephalans? As it looks in the phylogenetic structure in Figure 2, there seems to be a large clade with a mix of high (red) and low (blue) AEs (nematodes?), a mid-sized clade with predominantly low (blue) AEs (platyhelminths?), and a small clade with predominantly mid-high (red) AEs (acanthocephalans?).

Furthermore, the following minor comments should also be considered by the authors:

L18: “Helminth parasites are part of almost every ecosystem, with an estimated 1-2 billion people infected at any time.” I find this sentence a bit misleading. Although helminth parasites are indeed very diverse and are present in nearly every ecosystem, only very few taxa in this group are actually infecting humans. Likewise, L35 could be changed to “Some helminth species also take a heavy toll on human health [...]”

L57: Please add a comma after “reference temperature”

L101-102: “We thus focus our analyses on the thermal sensitivity of developmental processes from here on” AE could be mentioned here as well.

L112-143: These are all important factors that might determine the parasites’ temperature sensitivity. Additional factors that could be tested in future analyses could include ‘size of the parasite stage’, ‘host specificity’, ‘type of parasite stage’ (e.g., active transmission stage, active reproduction stage, passive waiting stage etc.), or a further distinction of the aquatic habitat type into ‘freshwater’ vs. ‘marine’. I am not suggesting to include these in the current manuscript but they could be used in future assessments.

L194: “aquatic habitats could buffer temperature fluctuations” While this is certainly true, it should be noted that many marine organisms (and their parasites) in coastal zones regularly experience dramatic temperature shifts during the tidal cycle (e.g., in small rock pools, or when falling completely dry during low tide).

L215: What does ‘eV’ stand for?

L248: insert space after “prokaryotes”

L249-251: “meaningful thermal performance data exist for less than 0.1% of the estimated hundreds of thousands of extant helminth species” Moreover, the available data seems skewed towards certain groups of parasites, since the current dataset consists largely of thermal performance values from nematodes, despite trematodes being estimated to make up the largest part of helminth diversity (see [54] Carlson 2020)

L306-307: “despite the greater temperature fluctuations experienced in terrestrial environments” See my comment regarding intertidal fluctuations.

L314: “Perhaps it is more beneficial to be able to react quickly to the onset of short growing seasons” ...especially since transmission windows can be expected to be shorter in cold regions.

Decision letter (RSPB-2021-1878.R0)

28-Sep-2021

Dear Ms Phillips:

Your manuscript has now been peer reviewed and the reviews have been assessed by an Associate Editor. The reviewers' comments (not including confidential comments to the Editor) and the comments from the Associate Editor are included at the end of this email for your reference. As you will see, the reviewers have raised some concerns with your manuscript and we would like to invite you to revise your manuscript to address them.

Research ethics:

Use of animals and field studies:

It is a condition of publication that you make available the data and research materials supporting the results in the article. Please see our Data Sharing Policies (<https://royalsociety.org/journals/authors/author-guidelines/#data>). Datasets should be deposited in an appropriate publicly available repository and details of the associated accession number, link or DOI to the datasets must be included in the Data Accessibility section of the

article (<https://royalsociety.org/journals/ethics-policies/data-sharing-mining/>). Reference(s) to datasets should also be included in the reference list of the article with DOIs (where available).

Please submit a copy of your revised paper within three weeks. If we do not hear from you within this time your manuscript will be rejected. If you are unable to meet this deadline please let us know as soon as possible, as we may be able to grant a short extension.

Best wishes,
Professor Hans Heesterbeek
mailto: proceedingsb@royalsociety.org

Associate Editor

Board Member: 1

Comments to Author:

Thank you for allowing Proc B to consider this extremely interesting and timely MS. As you'll read below, both reviewers were impressed and believe that the work will advance our understanding of parasite transmission in a changing climate. However, they also identified some important weaknesses that should be addressed. Please pay attention to the following suggestions in particular:

1. REVISE INTRO - Please bring the intro into alignment with the discussion by presenting the concept of AE comprehensively in the introduction section, as suggested by Reviewer 2.
2. QUANTITATIVE COMPARISON OF PARASITE CLADES - Please provide a quantitative comparison of AE among the major parasite clades (e.g., cestodes, trematodes, etc) as requested by Reviewer 2. This can be written in the results or presented as a table/figure at your discretion.

3. HOST PHYLOGENY AND BODY SIZE - Please consider Reviewer 1's suggestion about considering the effects of host phylogeny and body size on parasite AE. You may choose to include these variables in statistical models but, if you do not, please at least address the issue in the discussion.

Reviewer(s)' Comments to Author:

Referee: 1

Comments to the Author(s)

The effects of phylogeny, habitat, and host characteristics on the thermal sensitivity of helminth development

This topic is both very interesting and relevant, and I am happy to see these authors have compiled such a vast and important dataset for helminths. I think the stronger phylogenetic conservatism supported by their dataset is important (and surprising), as I initially thought parasite traits/ecology would be more important. Overall, the manuscript is well written, and the take home messages are clear.

I only have a few minor suggestions for the authors to consider:

1. I wonder if conducting a comparative analysis of temperature dependences with the addition of host phylogeny is worth considering. It seems like both parasite and host phylogeny would be important and can drive some macroecological patterns (e.g., Harnos et al., 2016 'Size matters for lice on birds: Coevolutionary allometry of host and parasite body size' *Evolution*). Thinking about this further, it could be problematic for multi host lifecycles, so I am unsure what host in the lifecycle is the most appropriate to model.

2. Within host groups (e.g., ectotherms), I suspect host body size could be an important covariate in the models presented here. The idea being bigger hosts can provide some sort of thermal inertia for parasites (similar to the ideas in Bergmann's Rule).

Line 292: I think this sentence should start 'The fact that phylogeny...'

Lines 313-317: This part of the discussion makes me wonder if temperature seasonality (coefficient of variation of temp) would be a useful covariate in the model.

Table 1. Is there a graphical way to represent this information? Maybe consider displaying the estimates of each term in the model. Further, showing the direction of the effects of the covariates would be useful.

Referee: 2

Comments to the Author(s)

The authors present a study on the thermal sensitivity of helminth parasites by comparing the activation energies (AEs) for developmental rates based on available data from experimental studies. To my knowledge, this study presents the first approach that analyses the thermal sensitivity of multiple parasite helminth taxa from different phyla in such a unifying framework. Ultimately, such large-scale assessments will be useful to better estimate how different parasite taxa will respond to climate change and global warming pressures. However, as the authors point out, empirical data gaps still hinder current assessments and predictions and will need to be addressed to provide better predictive frameworks for the impacts of climate change on parasite dynamics. Overall, this is a relevant and well-written manuscript that should be of interest to a wide readership.

My main criticism is that the introduction and discussion could be 'synchronised' a bit better. In particular, in the introduction the activation energy AE is only mentioned briefly in the context of the BA model (L58, and as an example in L69), yet throughout the rest of the text AE is used and

discussed as the main indicator of thermal sensitivity in helminth development. In my opinion, AE could therefore be explained a bit more in detail in the introduction for a general readership.

Secondly, the authors conclude that “phylogenetic structure was key for explaining the variation in thermal sensitivity in our dataset” (L267). Was there any overall difference between nematodes, trematodes, cestodes or acanthocephalans? As it looks in the phylogenetic structure in Figure 2, there seems to be a large clade with a mix of high (red) and low (blue) AEs (nematodes?), a mid-sized clade with predominantly low (blue) AEs (platyhelminths?), and a small clade with predominantly mid-high (red) AEs (acanthocephalans?).

Furthermore, the following minor comments should also be considered by the authors:

L18: “Helminth parasites are part of almost every ecosystem, with an estimated 1-2 billion people infected at any time.” I find this sentence a bit misleading. Although helminth parasites are indeed very diverse and are present in nearly every ecosystem, only very few taxa in this group are actually infecting humans. Likewise, L35 could be changed to “Some helminth species also take a heavy toll on human health [...]”

L57: Please add a comma after “reference temperature”

L101-102: “We thus focus our analyses on the thermal sensitivity of developmental processes from here on” AE could be mentioned here as well.

L112-143: These are all important factors that might determine the parasites’ temperature sensitivity. Additional factors that could be tested in future analyses could include ‘size of the parasite stage’, ‘host specificity’, ‘type of parasite stage’ (e.g., active transmission stage, active reproduction stage, passive waiting stage etc.), or a further distinction of the aquatic habitat type into ‘freshwater’ vs. ‘marine’. I am not suggesting to include these in the current manuscript but they could be used in future assessments.

L194: “aquatic habitats could buffer temperature fluctuations” While this is certainly true, it should be noted that many marine organisms (and their parasites) in coastal zones regularly experience dramatic temperature shifts during the tidal cycle (e.g., in small rock pools, or when falling completely dry during low tide).

L215: What does ‘eV’ stand for?

L248: insert space after “prokaryotes”

L249-251: “meaningful thermal performance data exist for less than 0.1% of the estimated hundreds of thousands of extant helminth species” Moreover, the available data seems skewed towards certain groups of parasites, since the current dataset consists largely of thermal performance values from nematodes, despite trematodes being estimated to make up the largest part of helminth diversity (see [54] Carlson 2020)

L306-307: “despite the greater temperature fluctuations experienced in terrestrial environments” See my comment regarding intertidal fluctuations.

L314: “Perhaps it is more beneficial to be able to react quickly to the onset of short growing seasons” ...especially since transmission windows can be expected to be shorter in cold regions.

Author's Response to Decision Letter for (RSPB-2021-1878.R0)

See Appendix A.

RSPB-2021-1878.R1 (Revision)

Review form: Reviewer 1

Recommendation

Accept as is

Scientific importance: Is the manuscript an original and important contribution to its field?

Excellent

General interest: Is the paper of sufficient general interest?

Good

Quality of the paper: Is the overall quality of the paper suitable?

Excellent

Is the length of the paper justified?

Yes

Should the paper be seen by a specialist statistical reviewer?

No

Do you have any concerns about statistical analyses in this paper? If so, please specify them explicitly in your report.

No

It is a condition of publication that authors make their supporting data, code and materials available - either as supplementary material or hosted in an external repository. Please rate, if applicable, the supporting data on the following criteria.

Is it accessible?

Yes

Is it clear?

Yes

Is it adequate?

Yes

Do you have any ethical concerns with this paper?

No

Comments to the Author

The authors have addressed the comments thoroughly. I have no further suggestions and believe this work will be important to the field.

Decision letter (RSPB-2021-1878.R1)

04-Jan-2022

Dear Ms Phillips

I am pleased to inform you that your manuscript entitled "The effects of phylogeny, habitat, and host characteristics on the thermal sensitivity of helminth development" has been accepted for publication in Proceedings B.

Data Accessibility section

Open Access

You are invited to opt for Open Access, making your freely available to all as soon as it is ready for publication under a CCBY licence. Our article processing charge for Open Access is £1700. Corresponding authors from member institutions (<http://royalsocietypublishing.org/site/librarians/allmembers.xhtml>) receive a 25% discount to these charges. For more information please visit <http://royalsocietypublishing.org/open-access>.

Your article has been estimated as being 9 pages long. Our Production Office will be able to confirm the exact length at proof stage.

Paper charges

Sincerely,

Professor Hans Heesterbeek

Associate Editor:

Board Member: 1

Comments to Author:

Thank you for your careful attention to the last round of revisions and for your thoughtful responses to reviewers. I am now satisfied that all the issues raised during peer review have been addressed. Congratulations on a lovely paper!

Board Member: 2

Comments to Author:

(There are no comments.)

Appendix A

Dear Professor Heesterbeek,

We are pleased to have the opportunity to submit a revised version of our manuscript, “The effects of phylogeny, habitat, and host characteristics on the thermal sensitivity of helminth development” for publication in the *Proceedings of the Royal Society of London Series B* (RSPB-2021-1878). None of the material presented in this paper is published elsewhere or is under consideration for publication elsewhere.

Below, we offer responses (see italicized font) to the suggestions and concerns of the Associate Editor and Reviewers on a point-by-point basis (note that line numbers refer to the revised manuscript with ‘simple markup’ unless otherwise indicated). Thank you again for your consideration of this manuscript.

Sincerely,

Jessica Ann Phillips, Juan S. Vargas Soto, Samraat Pawar, Janet Koprivnikar, Daniel P. Benesh, Péter K. Molnár

Associate Editor

Board Member: 1

Comments to Author:

Thank you for allowing Proc B to consider this extremely interesting and timely MS. As you’ll read below, both reviewers were impressed and believe that the work will advance our understanding of parasite transmission in a changing climate. However, they also identified some important weaknesses that should be addressed. Please pay attention to the following suggestions in particular:

1. REVISE INTRO - Please bring the intro into alignment with the discussion by presenting the concept of AE comprehensively in the introduction section, as suggested by Reviewer 2.

We have revised the Introduction as suggested (see Lines 50-67) and now provide more information regarding AEs.

2. QUANTITATIVE COMPARISON OF PARASITE CLADES - Please provide a quantitative comparison of AE among the major parasite clades (e.g., cestodes, trematodes, etc) as requested by Reviewer 2. This can be written in the results or presented as a table/figure at your discretion.

Please see our response to Reviewer 2 below. We have added a figure to the supplement that illustrates the negligible effect of phylum relative to family when considering comparisons of AE among the helminth taxa.

3. HOST PHYLOGENY AND BODY SIZE - Please consider Reviewer 1’s suggestion about considering the effects of host phylogeny and body size on parasite AE. You may choose to include these variables in statistical models but, if you do not, please at least address the issue in the discussion.

Please see our response to Reviewer 1 below. Specifically, we note that the addition of host body size and phylogeny would be challenging because the vast majority of helminth parasites have complex, multi-

host life cycles. As such, it would be necessary to decide which host(s) were relevant in this context (e.g. first or second intermediate hosts or definitive hosts). In addition, a substantial number of activation energy estimates in our analysis are for stages outside of a host, and choosing the host for a stage that occurs either before or after the free-living stage would be an arbitrary decision. It is thus not evident how one might incorporate a single appropriate value related to either host body size or phylogeny for each helminth species. In addition, host phylogeny is already encompassed by some of our covariates, such as the use of animal or plant hosts or aquatic vs. terrestrial life cycles.

We now address this in the Discussion on lines 353-356:” We did not include host phylogeny or body size in this analysis because many helminths have complex life cycles within multi-host systems, and most of our data are restricted to the free-living life stages of parasites. This reflects the difficulty of experimentally measuring traits of helminth parasites across multiple life stages, and especially when they are within hosts.”

Reviewer(s)' Comments to Author:

Referee: 1

Comments to the Author(s)

The effects of phylogeny, habitat, and host characteristics on the thermal sensitivity of helminth development

This topic is both very interesting and relevant, and I am happy to see these authors have compiled such a vast and important dataset for helminths. I think the stronger phylogenetic conservatism supported by their dataset is important (and surprising), as I initially thought parasite traits/ecology would be more important. Overall, the manuscript is well written, and the take home messages are clear.

Thank you!

I only have a few minor suggestions for the authors to consider:

1. I wonder if conducting a comparative analysis of temperature dependences with the addition of host phylogeny is worth considering. It seems like both parasite and host phylogeny would be important and can drive some macroecological patterns (e.g., Harnos et al., 2016' Size matters for lice on birds: Coevolutionary allometry of host and parasite body size' Evolution). Thinking about this further, it could be problematic for multi host lifecycles, so I am unsure what host in the lifecycle is the most appropriate to model.

We agree that host phylogeny can be important when investigating certain macroecological patterns, specifically when considering influences of parasitism on hosts. However, our analyses are focused on explaining patterns of AE in parasites, rather than their hosts, and many host traits, including phylogeny, are comparatively less relevant in this context, such as for free-living parasite propagule stages. Some host traits may have an influence on parasite AEs, for example, via the temperature sensitivity of the survival of ectotherm intermediate hosts and the resultant time window that is available for parasite development. However, accounting for such effects through host phylogeny is not possible given the

currently available data, both because related parasites tend to use related hosts (thus confounding parasite and host phylogeny) and because many helminths have complex life cycles, but thermal data only exists for one parasite life stage and often only the free-living ones, as the Reviewer also mentions. Having said that, our analyses do account implicitly for some potential variation through host phylogeny through many of our fixed effects: namely use of plants or animals as hosts, and considering terrestrial vs. aquatic life cycles.

We have now addressed this in the Discussion on lines 353-356.

2. Within host groups (e.g., ectotherms), I suspect host body size could be an important covariate in the models presented here. The idea being bigger hosts can provide some sort of thermal inertia for parasites (similar to the ideas in Bergmann's Rule).

As mentioned in our preceding response, there is no obvious way to choose which host to focus upon for free-living parasite stages, which encompass a large portion of our dataset. For example, there is no host size for egg development, and selecting a host for this stage would be an arbitrary decision. It isn't evident how to appropriately calculate or include host body size in these cases to obtain a single measure for each helminth species.

We have now addressed this in the Discussion on lines 353-356.

Line 292: I think this sentence should start 'The fact that phylogeny...'

We have modified this sentence to start with "The fact that phylogeny..." on line 294.

Lines 313-317: This part of the discussion makes me wonder if temperature seasonality (coefficient of variation of temp) would be a useful covariate in the model.

We did not include the coefficient of temperature variation in our models, but we did include three variables likely correlated with temperature fluctuations: the range in monthly temperature means (Fig. 3b), latitude (Fig. 3c), and habitat (Fig. 3d). AE was not strongly correlated with the first two, but AE did tend to be higher in terrestrial vs aquatic parasites (lines 243-244). Testing whether this reflects adaptation to temperature variability, such as through finer measurements of temperature across seasons, could be an interesting avenue for future research, which we discuss from lines 313-320.

Table 1. Is there a graphical way to represent this information? Maybe consider displaying the estimates of each term in the model. Further, showing the direction of the effects of the covariates would be useful.

The goal of Table 1 is to present the model-building approach and summarize model fits, whereas the effect sizes and directions for many model terms are plotted in Figures 3 and 4, and numerous effect sizes, such as slopes, are stated in the results. Thus, while we agree that graphical representations can be helpful, we think that doing so, in this case, would introduce some redundancy.

Referee: 2

Comments to the Author(s)

The authors present a study on the thermal sensitivity of helminth parasites by comparing the activation energies (AEs) for developmental rates based on available data from experimental studies. To my knowledge, this study presents the first approach that analyses the thermal sensitivity of multiple parasite helminth taxa from different phyla in such a unifying framework. Ultimately, such large-scale assessments will be useful to better estimate how different parasite taxa will respond to climate change and global warming pressures. However, as the authors point out, empirical data gaps still hinder current assessments and predictions and will need to be addressed to provide better predictive frameworks for the impacts of climate change on parasite dynamics. Overall, this is a relevant and well-written manuscript that should be of interest to a wide readership.

Thank you!

My main criticism is that the introduction and discussion could be 'synchronised' a bit better. In particular, in the introduction the activation energy AE is only mentioned briefly in the context of the BA model (L58, and as an example in L69), yet throughout the rest of the text AE is used and discussed as the main indicator of thermal sensitivity in helminth development. In my opinion, AE could therefore be explained a bit more in detail in the introduction for a general readership.

Thank you for this suggestion. We have revised the Introduction to clarify and provide additional information regarding the meaning and interpretation of AE (see Lines 50-67).

Secondly, the authors conclude that "phylogenetic structure was key for explaining the variation in thermal sensitivity in our dataset" (L267). Was there any overall difference between nematodes, trematodes, cestodes or acanthocephalans? As it looks in the phylogenetic structure in Figure 2, there seems to be a large clade with a mix of high (red) and low (blue) AEs (nematodes?), a mid-sized clade with predominantly low (blue) AEs (platyhelminths?), and a small clade with predominantly mid-high (red) AEs (acanthocephalans?).

This is a fair question. Phylogenetic structure can arise anywhere in the tree (i.e. near the tips or near the root), and although Figure 2 gives a sense of where there are shifts in activation energy (AE), it can still be hard to compare clades, like nematodes vs cestodes. Here is a box plot comparing the three helminth phyla in our dataset:

As the reviewer suggests, it looks like Platyhelminthes have lower AE than nematodes, on average. However, we believe it would be premature to conclude that helminth phyla differ in temperature sensitivity because these differences are at least partly driven by some overrepresented taxa. For instance, the median AE for Platyhelminthes is lower because one large taxon, tapeworms in the family Hymenolepidae (58% of the platyhelminth thermal performance curves), happen to have low AE. Other platyhelminth families had AEs closer to the overall median (Fig. S2 in the original submission, now Fig. S3).

The family-level means in Fig. S3 were estimated with a model including all taxonomic levels (phyla to species), i.e. the taxonomic version of the phylogenetic ‘base’ model in Table 1. We had noted in the supplement that family was the most important taxonomic level in this model. A complementary way to examine where differences in temperature sensitivity arise is to fit a series of nested models. Specifically, we added taxonomic levels to the model from the root to the tips, i.e. start with a model with just phylum, then add class, then order, etc. A given taxonomic level should improve the model if AE differs between groups and is consistent within them. At each step, we quantified the variation explained by taxonomy. The largest jumps in explanatory power were observed when we added order and family to the model; adding order increased the variation explained from 17 to 46%, and family increased it further to 59%. This is summarized in the following figure, which is now Figure S2 in the supplement:

Note that although the phylum-only model explains little variation in AE, there are wide credible intervals around the phylum-level random effect. Thus, the available data suggest that helminth phyla do not differ in temperature sensitivity overall, but given the limited number of species studied to date, further sampling is needed to strengthen this conclusion. We hope that our analysis, by identifying which helminth clades have and have not been studied, will stimulate such research.

In sum, we made the following changes to better illustrate where there are differences in temperature sensitivity: (i) we added phyla labels to Fig. 2 (e.g., “Nematoda” on the branch leading to nematodes), (ii) we noted in the results section that differences were mainly among orders and families (line 233-234), and (iii) we added to the supplement the analysis in which taxonomic levels were added sequentially, highlighting orders and families as most explanatory (see section 1.3 in supplement S1).

Furthermore, the following minor comments should also be considered by the authors:

L18: “Helminth parasites are part of almost every ecosystem, with an estimated 1-2 billion people infected at any time.” I find this sentence a bit misleading. Although helminth parasites are indeed very diverse and are present in nearly every ecosystem, only very few taxa in this group are actually infecting humans. Likewise, L35 could be changed to “Some helminth species also take a heavy toll on human health [...]”

We have modified the sentence in line 18 to “Helminth parasites are part of almost every ecosystem, with more than 300,000 species worldwide.” We also modified the sentence in lines 35-36 to “Some helminth species also take a heavy toll on human health, such as schistosomiasis or ascariasis.” as the reviewer suggested.

L57: Please add a comma after “reference temperature”
We have now added this comma on line 58.

L101-102: “We thus focus our analyses on the thermal sensitivity of developmental processes from here on” AE could be mentioned here as well.

We have revised this sentence to “We thus focus our analyses on the activation energies of developmental processes from here on.” on lines 104-105.

L112-143: These are all important factors that might determine the parasites’ temperature sensitivity. Additional factors that could be tested in future analyses could include ‘size of the parasite stage’, ‘host specificity’, ‘type of parasite stage’ (e.g., active transmission stage, active reproduction stage, passive waiting stage etc.), or a further distinction of the aquatic habitat type into ‘freshwater’ vs. ‘marine’. I am not suggesting to include these in the current manuscript but they could be used in future assessments.

We now note in the Discussion (lines 356-360) that such potential influences should be considered in future investigations.

L194: “aquatic habitats could buffer temperature fluctuations” While this is certainly true, it should be noted that many marine organisms (and their parasites) in coastal zones regularly experience dramatic temperature shifts during the tidal cycle (e.g., in small rock pools, or when falling completely dry during low tide).

This sentence gave the rationale for testing a latitude by habitat interaction. Specifically, seasonal changes in temperature, approximated by latitude, may differ in terrestrial and aquatic habitats and thus shape AE. We have modified this sentence to clarify that this term relates to seasonal temperature fluctuations (lines 193-195). On a daily time scale, though, some aquatic habitats, like tide pools, can experience pronounced temperature swings. We have acknowledged this in the Discussion (line 308-309 and see comment below).

L215: What does ‘eV’ stand for?

It stands for electron volts. We have now clarified this in the text on lines 217-218:” The median AE was 0.67 electron volts (eV)...”

L248: insert space after “prokaryotes”

Done (line 251).

L249-251: “meaningful thermal performance data exist for less than 0.1% of the estimated hundreds of thousands of extant helminth species” Moreover, the available data seems skewed towards certain groups of parasites, since the current dataset consists largely of thermal performance values from nematodes, despite trematodes being estimated to make up the largest part of helminth diversity (see [54] Carlson 2020)

We agree that such bias is a problem, and this is noted on line 336: “Our study is limited by the species that have been studied...” We have also added a sentence on lines 348-349: “Nematodes, for example, are overrepresented in our datasets, despite trematodes estimated to represent the largest part of helminth diversity (54).”

L306-307: “despite the greater temperature fluctuations experienced in terrestrial environments” See my comment regarding intertidal fluctuations.

In general, we believe it is a fair assumption that air and soil heat up and cool down faster than water, so terrestrial habitats should, on average, exhibit more temperature variability than aquatic ones. Nonetheless, the reviewer is correct that some aquatic habitats like intertidal and ephemeral ponds are also characterized by large temperature swings. We have revised the sentence to acknowledge this (line 307-309): “Furthermore, AE was higher in terrestrial than in aquatic helminths, despite presumably greater temperature fluctuations in terrestrial environments (though some aquatic habitats, such as tide pools, may also exhibit high temperature variability).”

L314: “Perhaps it is more beneficial to be able to react quickly to the onset of short growing seasons ...especially since transmission windows can be expected to be shorter in cold regions.

We have revised this sentence to “Perhaps it is more beneficial to be able to react quickly to the onset of the shorter growing seasons found in colder regions (high AEs), than to be able to buffer large temperature fluctuations during the growing season with low AEs.” on lines 316-319.